# Considerations for Home-Based Treatment of Fabry Disease in Poland during the COVID-19 Pandemic and Beyond

**DOI:** 10.3390/ijerph18168242

**Published:** 2021-08-04

**Authors:** Michał Nowicki, Stanisława Bazan-Socha, Mariusz Kłopotowski, Beata Błażejewska-Hyżorek, Mariusz Kusztal, Krzysztof Pawlaczyk, Jarosław Sławek, Andrzej Oko, Zofia Oko-Sarnowska

**Affiliations:** 1Department of Nephrology, Hypertension, and Kidney Transplantation, Medical University of Łódź, 90-145 Łódź, Łódzkie, Poland; 2Department of Internal Medicine, Jagiellonian University Medical College, 30-688 Kraków, Małopolskie, Poland; mmsocha@cyf-kr.edu.pl; 3Department of Interventional Cardiology and Angiology, Institute of Cardiology, 04-628 Warsaw, Mazowieckie, Poland; mklopotowski@kard.pl; 42nd Department of Neurology, Institute of Psychiatry and Neurology, 02-957 Warsaw, Mazowieckie, Poland; hyzorek@ipin.edu.pl; 5Department of Nephrology and Transplantation Medicine, Wrocław Medical University, 50-556 Wrocław, Dolnośląskie, Poland; mariusz.kusztal@umed.wroc.pl; 6Department of Nephrology, University Hospital of Karol Marcinkowski in Zielona Góra, 65-046 Zielona Góra, Lubuskie, Poland; kpawlac@ump.edu.pl; 7Department of Nephrology, Transplantology, and Internal Diseases, Poznań University of Medical Sciences, 60-355 Poznań, Wielkopolskie, Poland; aoko@ump.edu.pl; 8Division of Neurological and Psychiatric Nursing, Institute of Nursing and Midwifery, Medical University of Gdańsk, 80-462 Gdańsk, Pomorskie, Poland; jaroslawek@gumed.edu.pl; 9Department of Cardiology, Poznań University of Medical Sciences, 60-355 Poznań, Wielkopolskie, Poland; zokosar@wp.pl

**Keywords:** enzyme replacement therapy, Fabry disease, home-based treatment, telemedicine

## Abstract

Current therapy for Anderson–Fabry disease in Poland includes hospital or clinic-based intravenous enzyme replacement therapy with recombinant agalsidase alpha or beta, or oral pharmacological chaperone therapy with migalastat. Some countries around the world offer such treatment to patients in the comfort of their own homes. The 2020–2021 COVID-19 pandemic has pushed global healthcare providers to evolve their services so as to minimize the risk of COVID-19 exposure to both patients and providers; this has led to advances in telemedicine services and the increasing availability of at-home treatment for various procedures including parenteral drug administration. A total of 80% of surveyed Anderson–Fabry disease patients in Poland would prefer home-based treatment, which would be a safe and convenient alternative to clinic-based treatment if patient selection is based on our proposed algorithm. Our recommendations for home-based treatments appear feasible for the long term care of Anderson–Fabry disease patients during the COVID-19 pandemic and beyond. This may also serve as a basis for home-based treatment programs in other rare and ultra-rare genetic diseases.

## 1. Introduction

Anderson–Fabry disease (FD) is a rare X-linked lysosomal storage disorder caused by pathogenic mutations of the alpha-galactosidase A encoding gene [1]. Alpha-galactosidase A deficiency causes globotriaosylceramide and related glycosphingolipid accumulation. The classic phenotype presents in childhood and progresses, culminating in cardiac, cerebrovascular, and end-stage renal disease later in life. The highly variable clinical manifestations of FD mean its diagnosis is often significantly delayed or missed [2].

Current therapy for FD includes intravenously administered enzyme replacement therapy (ERT) with recombinant agalsidase alpha or beta, or oral pharmacological chaperone therapy with migalastat in patients with amenable mutations. Adjunct therapies include lifestyle modifications and prophylactic medications.

ERT for FD has been available since 2001 and is associated with important benefits that preserve critical organ function, including the stabilization of renal function and prevention of end-stage renal disease, an improvement in cardiac structure and function, a reduction in neuropathic pain severity, and an improvement in gastrointestinal symptoms. ERT requires intermittent intravenous drug infusions performed mostly in hospital or designated outpatient settings. Although possible for eligible patients in many countries, home treatment for FD is not practiced in Poland. Patients’ attitudes towards home treatment are expected to change significantly due to the current Coronavirus Disease-19 (COVID-19) pandemic and prolonged worldwide health crisis.

The 2020–2021 COVID-19 pandemic has dramatically changed how both hospital and outpatient care is delivered [3]. The unfolding pandemic is transforming medical care and medical personnel–patient communication more than any other crisis in modern history. When possible, in-person visits are converted to telemedicine appointments, and procedures including parenteral drug administration are increasingly being conducted at home. Many patients are avoiding visits to medical centers to minimize their risk of infection. Health provider behaviors are also changing in response to ever-evolving local and government restrictions on travel and non-essential services.

Years 2020 and 2021 thus far have witnessed a rapid increase in the use of telehealth technology for remote medical appointments, with this trend likely to continue into the coming years [4]. The pandemic has accelerated telemedicine, which became an effective solution for providing safe patient care in most medical cases, since it allows immediate healthcare access while minimizing in-person contact.

This position statement on the ERT for FD home infusion program in Poland was written by the multidisciplinary group of expert physicians directly involved in managing lysosomal storage disease patients, in response to an urgent and unmet need during the COVID-19 pandemic for a home-based alternative to the existing in-center drug infusion program.

## 2. Current Fabry Disease Patient Care Systems in Poland and Other EU Countries

At the end of 2019, the first FD patients in Poland received reimbursed ERT after a national drug program was launched by the Polish National Health Fund. At the end of 2020, oral chaperone therapy migalastat was made available through the same program for the subpopulation of FD patients with amenable mutations. However, the drug prescriptions covered by the program are subject to multiple restrictions and are not fully compatible with comparable programs in other EU countries. There is no centralized process for preparing therapeutic guidelines for rare and ultra-rare diseases including FD in Poland. The first nationwide recommendations for FD treatment were published by our group in the Polish Archives of Internal Medicine in early 2020 [5]. Our recommendations were based on medical literature analysis and recommendations from medical expert groups in other countries; therefore, the indications and contraindications for the therapy were not fully compatible with the drug reimbursement program (available at https://www.gov.pl/web/zdrowie/choroby-nieonkologiczne) (accessed on 23 June 2021).

Currently, 23 Polish medical centers have signed a contract with the Polish National Health Fund and offer FD pharmacotherapy to approximately 50 patients. These centers are unevenly distributed across the country, located in only 11 of the 16 voivodships. Most centers are in local hospitals or outpatient facilities, and each treats only 1–2 patients living close by. The remaining sites are larger academic centers, each treating 3–12 patients. To fulfill the drug program requirements, the centers must provide complex medical care including diagnostic tests (performed in their own laboratories or an external facility), medical procedures required to qualify patients for the initiation and continuation of therapy, and facilities for ERT or migalastat administration. Each FD patient’s commencement and continuation of reimbursed therapy must be approved by the single, designated central commission. 

Within Europe, there is considerable variation in how FD care and therapy are provided [6]. In the United Kingdom, Germany and France, the general principles and therapeutic systems are similar to those in Poland. They feature several nationally designated centers that prescribe therapy and provide genetic counselling, diagnostics, clinical assessments, and ongoing supportive care for those requiring specific therapy such as ERT, and those requiring follow-up. A lead physician and nurse team specialized in FD treatment coordinate multidisciplinary patient management. Some countries such as Germany, the Netherlands, and Italy also provide more local ERT administration, including at-home services [7,8,9,10]. 

Some smaller countries, including Slovenia, Czechia, Slovakia, Lithuania, and Hungary, have centralized FD patient care systems, with a single specialized national center in each country. These centers not only offer complex multidisciplinary care for FD patients but also educational activities for medical professionals [11]. The physicians specializing in FD patient care consider this centralized system to be effective, with highly satisfactory access to diagnostic tests and therapy, but such an approach might be not be suitable for larger countries, including Poland. Some of these countries, e.g., Slovakia, also offer home infusions. 

A centralized approach with regional specialization of services requires most patients to travel long distances for each medication infusion; therefore, this solution is preferable only in smaller countries. Many patients find frequent hospital visits stressful, time consuming, expensive, and disruptive, particularly during the COVID-19 pandemic. Home therapy can be conveniently integrated into a patient’s daily life, and may also reduce the utilization of hospital resources and total care costs. The rapid spread of COVID-19 has brought about numerous changes to healthcare systems to minimize the infection risk for staff and patients, particularly as FD patients are at greater risk of COVID-19 morbidity and mortality. We believe that home-based ERT therapy, together with the broader use of telemedicine resources, would be advantageous for FD patients in Poland. 

Our group members recently collected important information on the expectations and fears of FD patients receiving ERT in Poland during the COVID-19 pandemic [12]. The anonymous customized survey was distributed to all FD patients visiting Polish FD centers between 20 September and 1 November 2020. Importantly, 80% of surveyed patients would prefer home infusions provided by a nurse or trained non-medical person, whereas only 14.5% patients would prefer ERT treatment in the local hospital or medical center where they are currently treated. Patients identified long and costly transportation to the medical center as the most significant obstacle in receiving their current in-center ERT. They also indicated that they were very afraid of the risk of exposure to COVID-19 infection enroute to and while at the medical center for their infusion. The survey results provide a strong argument for the availability of home ERT infusions in Poland as an alternative to the current in-center treatments [12].

## 3. Home-Based Therapies in Other Countries

Home-based infusions for FD patients are provided in many countries around the world, as shown in Table 1; they have been demonstrated as safe and effective in such countries as Italy [7,8], the Netherlands [13], Norway [14], Germany, Austria [15], the USA [16], and Argentina [17]. The early experience reported in the literature on the impact of COVID-19 on lysosomal storage disorder therapies showed that 49% of patients receiving hospital-based ERT experienced treatment disruption [18]. 

The Norwegian government is known for striving for a high quality of life for all its citizens, including people with rare diseases and their families. There are a number of specialized centers for rare diseases in Norway, including two for FD, i.e., one each in Oslo and Bergen [14]. Encouraging more patients to undergo home-based infusions has the potential to free-up community-based resources and improve patients’ quality of life [14]. Successful organization and patient education regarding home-based ERT for FD could serve as a model for home-based infusions for other diseases in Norway [14]. 

COVID-19, caused by the SARS-CoV-2 virus, was declared a global pandemic by the World Health Organization in March 2020. Rare diseases, such as congenital errors of metabolism, were included in the group of conditions considered to have an extremely high risk of severe illness [8,24]. There are published reports providing practical recommendations related to ERT for FD during this period. Due to this lack of data, the analogy of recommendations for other congenital diseases is used to derive those for FD [8,24]. 

In Italy, for example, the home infusion program Fabry@Home has been in use since 2008 [7], and features a treating physician and registered nurse team dedicated to at-home ERT infusions. In 2017, Concolino et al. [7] reported results of a multi-center longitudinal study including more than 4000 home-delivered ERT infusions in a large cohort of Italian FD patients. They demonstrated that home-based treatment is as safe and effective as that at a hospital, but with improved treatment adherence and overall patient quality of life. Observational data from a large FD referral center in Naples also indicate that at-home care is the best solution for both health system resources and FD patients during the COVID-19 pandemic [8]. The Fabry Center of Federico II University of Naples is one of the main FD referral centers in Italy, with more than 150 patients, performing approximately 500 outpatient clinic visits annually [8]. No interruption or modification of treatment occurred for patients receiving oral therapy [8]. All patients receiving intravenous treatments were on home therapy and continued their infusions regularly, except eight patients (11%) who missed one infusion due to either flu-like symptoms or fear of infection.

Recommencing ERT after a treatment interruption does not fully reverse the clinical decline resulting from the disruption [24]. At-home treatment with correct use of personal protective equipment seems to be the most effective way to maintain access to therapy during the COVID-19 pandemic. If home therapy is unavailable, safe locations with strict separation of COVID-19 and non-COVID-19 patients should be provided in hospitals and infusion centers [24]. As seen in Czechia, alternative locations for ERT administration are dialysis centers, where the staff are familiar with renal patients and have relevant skills, such as gaining peripheral vasculature access. The network of dialysis units is so dense that patients can usually access one within a 50 km distance from home.

## 4. Telemedicine in FD Treatment and the Influence of the COVID-19 Pandemic

The growing popularity of virtual healthcare services over the past decade highlights the growing trend of connecting patients with the world of digital health information via smartphones and other mobile devices. Table 2 highlights basic mobile app functions that allow the documentation of a patient’s treatment, including vital signs and clinical photographs, infusion logs, alerts, and a home inventory of medications. 

The terms “telemedicine”, “telehealth” and “eHealth” refer to the use of electronic information and telecommunication technology to support and promote long-distance clinical healthcare, health-related education for patients and professionals, public health and healthcare administration [25]. Broader definitions include “telemedicine intervention” or “telehealth intervention” as the use of telecommunications technology to facilitate the remote delivery of healthcare services and clinical information [25,26]. These interventions can be synchronous or asynchronous, and include any information technology-based strategies for connecting healthcare professionals and patients through video conferencing, e-mail, remote electronic monitoring equipment, social network apps, and internet portals [27]. Telehealth interventions include interactive telemedicine services that facilitate concurrent interactions between patients, caregivers, and clinicians; they also allow remote monitoring of a patient’s health status using telehealth equipment and “store-and-forward telemedicine”, which transmits disease-related data such as medical images and biological measures [26]. The successful application of telemedicine has been reported in many chronic diseases such as diabetes, heart failure, asthma, COPD, hemophilia and lysosomal storage diseases [18,28]. 

Well organized home-therapy has been a highly efficient way to maintain treatment access during the COVID-19 pandemic in many European countries. Telemedicine, including both video and telephone communications, is emerging as an important tool for maintaining outpatient care while limiting direct patient contact [29]. An Italian example of at-home agalsidase infusion during the COVID-19 pandemic highlighted the applicability of telemedicine in FD patient care [8]; telemedicine was successfully adopted in the biggest center for rare diseases in Naples for all patients with FD, and none of the 129 interviewed patients acquired COVID-19 infection. This absence of infection may be attributed to the particular attention these patients paid to respecting preventative hygiene measures, and to the existing home-therapy programs, well-practiced in using telemedicine to monitor patients. In this system, the patient was contacted 24 to 48 h before their scheduled appointment by a clinic coordinator to confirm the visit date and time, and explain that it would take place in their own home using telehealth facilities. Medical staff called the patient at the scheduled appointment time, and asked them to measure their own temperature, pulse, blood pressure, and body weight. If necessary, visual examinations were performed, including respiratory function evaluations, using platforms such as FaceTime, WhatsApp, MS Teams, Google Meet, and Zoom. During the pandemic, laboratory tests were taken at a non-hospital-based laboratory. Medications and prescriptions could also be evaluated and adjusted remotely. Dried blood spot tests used in many countries to diagnose Fabry disease can also be used for disease monitoring because the lyso-GB-3 concentration can be assessed. 

Implementing telehealth programs in areas with poor access to health care facilities could expand patient access while reducing burdens such as the travel required to receive specialty care, and improving the monitoring, timeliness, and communication within the care continuum.

## 5. Basic Principles and Recommendations for the Polish FD Home Care Program

The basic principles of the FD home care program, including patient enrollment, monitoring and therapy evaluation, must follow the requirements of the drug program established in 2019 in Poland. We recommend some additional changes to the program that would accommodate potential at-home ERT administration; our recommendations appear feasible for the long-term care of FD patients both during and after the COVID-19 pandemic.

Patients will be enrolled for intravenous ERT or oral migalastat therapy in one of the specialized medical centers in contract with the Polish National Health Fund. Currently, each patient visits a center every other week for ERT administration, and every 6 months for a general health examination and assessment for ongoing therapy, evaluated by a central commission. We recommend that in certain situations such as the current pandemic, the assessments may be delayed if the patient’s health is assessed as stable by the local center, but should not occur less frequently than every 12 months. Additionally, every 3 months the patient should undergo a telemedicine appointment, e.g., by telephone or an internet-enabled device, at which time the lead doctor decides whether therapy should be continued, or if the patient should be seen at the center for an in-person health check-up. 

Individuals treated with migalastat should receive medical supplies every 3–6 months, depending on the lead physician’s decision. Patients receiving ERT, particularly during the COVID-19 pandemic, should have the opportunity to receive ERT at home. In agreement with recently published observational studies, we conclude that such an approach is as safe and effective as hospital administration. Suitability for home therapy would be determined according to each individual patient’s preferences, and by their eligibility as assessed by the lead FD center physician. 

To be eligible for the home ERT program, each FD patient must: have received at least five in-center ERT infusions without any evidence of severe adverse reactions during the last three infusions; be in a stable clinical condition, i.e., without any deterioration of target organ damage, such as renal, cardiovascular, and cerebrovascular; have given signed, informed consent before joining the program. 

Figure 1 gives details of the patient selection process for the initiation and continuation of at-home ERT.

The proposed FD home therapy program will involve a professional team consisting of a lead physician from an FD medical center and a registered nurse trained in FD and ERT. Patients on ERT should receive drug stock every 3 months, which will be provided by the FD medical center and received personally or delivered by the nurse directly at home. In collaboration with a hospital pharmacy team, a monitoring protocol should be implemented to ensure that the drug to be infused has been stored in controlled conditions and prepared adequately [30]. The drug is transported in a special thermostable container provided by the center. The nurse will visit patients every 2 weeks to perform an ERT infusion. The nurse is responsible for checking the drug vial condition before administration, recording details of administered vials (number of vials and batch numbers), and monitoring vital signs before and after administration and any adverse reactions during or after treatment. The nurse is present to monitor the patient’s health condition for at least 1 h after the infusion and record post-infusion vital signs. Assisted by a nurse, each patient will keep a diary of their symptoms and treatment record. During each infusion, the patient or patient’s caregiver must complete a general health/quality of life evaluation form using a visual analogue scale from 0 to 10, with 10 being “best imaginable health state” and 0 being “worst imaginable health state”; this information provides a quantitative measure of health outcome as judged by the individual respondents. Missing no more than five infusions per year will be acceptable.

Select patients may utilize a central vein port, but it is not a requirement for at-home ERT. 

Throughout the infusion, the nurse will remain in constant, real-time remote contact with the patient’s general practitioner, who will be notified immediately of any adverse event, for which they will then provide assistance. If required, the nurse can also contact (e.g., by telephone) one of the assistant physicians from the FD medical center. 

Infusion reactions will be managed by slowing or stopping the infusion and administering intravenous hydrocortisone or methylprednisolone, clemastine, intravenous fluids and inhaled salbutamol. Standard premedication will not be given, but any patient who experiences an infusion-related adverse event will be evaluated by the lead FD center physician for premedication with antihistamines and/or corticosteroids before subsequent infusions. Following an infusion-related adverse event, we recommend that the next three ERT administrations take place in a hospital or FD medical center. 

Infusion-associated events, such as fever, chills, or other reactions that occur during or within 4 h post-ERT, require anti-agalsidase IgG antibody measurement. A positive result does not exclude further at-home treatment, but close surveillance and premedication for subsequent infusions will be required. Patients who develop hypotension or dyspnea will require IgE investigation to rule out an anaphylactic reaction to the treatment product. A positive result precludes further at-home ERT administration. 

If a patient suffers general health deterioration, they can directly contact the general practitioner and/or assistant FD medical center physician using telemedicine technology. 

The patient, their treating physician, or home service nurse can submit a request to terminate the patient’s home therapy, and resume hospital-based ERT administration.

## 6. Adverse Effects of ERT

The most important factor influencing the choice between hospital-based and home-based therapy is the adverse effect profile of ERT in FD and other lysosomal storage diseases that require repeated intravenous infusions of recombinant proteins (enzymes). Adverse effects of agalsidase alfa or agalsidase beta IV infusions can be divided into two categories: infusion-associated reactions (IARs) and drug-related adverse effects (AEs). IARs occur during the infusion and up to 4 h after ERT administration. IARs in ERT patients must be distinguished from individuals with a humoral immunological response to infused proteins, especially in men with an absence of endogenous enzyme [13,31,32,33]. Any symptom appearing later than 4 h post-ERT but before the next infusion qualifies as an AE [13]. Literature reviews reveal some discrepancy between clinical, multi-center, and single-center studies. The frequency of IARs reported in the randomized studies was much higher (59–100%) than in observational studies (2.4–52%). According to the manufacturer’s product information, 24% of patients receiving agalsidase alfa in a recommended dose of 0.2 mg/kg developed IgG class antibodies vs. 67% of those receiving agalsidase beta in a dose of 1.0 mg/kg, but IARs occurred rarely [31]. This difference may be related to the five-times higher prescribed dose of agalsidase beta than agalsidase alfa [31,32]. The IARs mainly consisted of rigors, chills and fever. IARs are sufficiently managed with a prolonged infusion time, and in some cases require premedication with antihistamines and/or steroids [7,16,17,34,35,36,37,38,39]. Chest pain, tachycardia, hypertension, headache, nausea/vomiting, urticaria and erythema were among the demonstrated AEs. Severe, life-threatening events (SAEs) are reported as very rare complications of ERT [34,40,41]. This report and the finding that almost all patients were IgE and skin test negative suggested that the mechanism of the complications was different to IgE-mediated type-1 hypersensitivity [31]. Only 1.7–9.0% of patients treated with agalsidase beta and none receiving agalsidase alfa were IgE positive [16,17,33,34,35,36,37,38,41]. Other SAEs such as stroke or TIA, coronary artery disease, and sudden cardiac death were classified as adverse events of FD independently of ERT administration [16,34,36]. A significant ERT side-effect in FD patients is the formation of neutralizing antidrug antibodies (ADAs), which can limit treatment effectiveness [33]. These antibodies are predominantly IgG4 subtypes, and are detected in up to 70% of male patients in some studies [42,43]. They can bind to the infused recombinant enzyme, increase plasma globotriaosylceramide (Gb3) accumulation, and aggravate the disease course. The risk of ADA formation is very low in heterozygous females, related to the endogenous enzyme level [33,34,36,38,41], and may be higher in patients receiving agalsidase beta compared to those receiving agalsidase alfa (OR 2.8; *p* = 0.04) [42]. ADA-positive patients were older and had more advanced disease [44]. Increased ADA formation was observed in patients treated with agalsidase beta, and could be associated with the higher recommended doses, i.e., 1.0 mg/kg vs. 0.2 mg/kg for agalsidase alpha. During the follow-up period, lysoGb3 decreased in both drug groups, more so in patients without ADAs. In ADA-positive patients, serum lysoGb3 was higher and remained elevated compared to ADA-negative patients, while urinary lysoGb3 levels did not differ from baseline in the ADA-positive group.

## 7. Home-Based ERT for Fabry Disease in Poland—Challenges and Opportunities

This position statement written by a group of experts representing a range of clinical specialties may provide a basis for the establishment of a home-based ERT treatment program for FD and other rare diseases that require intermittent IV drug infusions. The authors showed that at-home therapy for FD is feasible, meets the patients’ currently unmet needs particularly during a pandemic, and would be safe if patients eligible for home therapy are carefully selected based on the proposed algorithm. The patients would be effectively monitored by qualified and experienced medical staff using remote-contact systems to ensure appropriate therapy management and minimal risk of potential infection. The experts’ considerations for successful home-based enzyme replacement therapy program implementation are summarized in Table 3.

## Figures and Tables

**Figure 1 ijerph-18-08242-f001:**
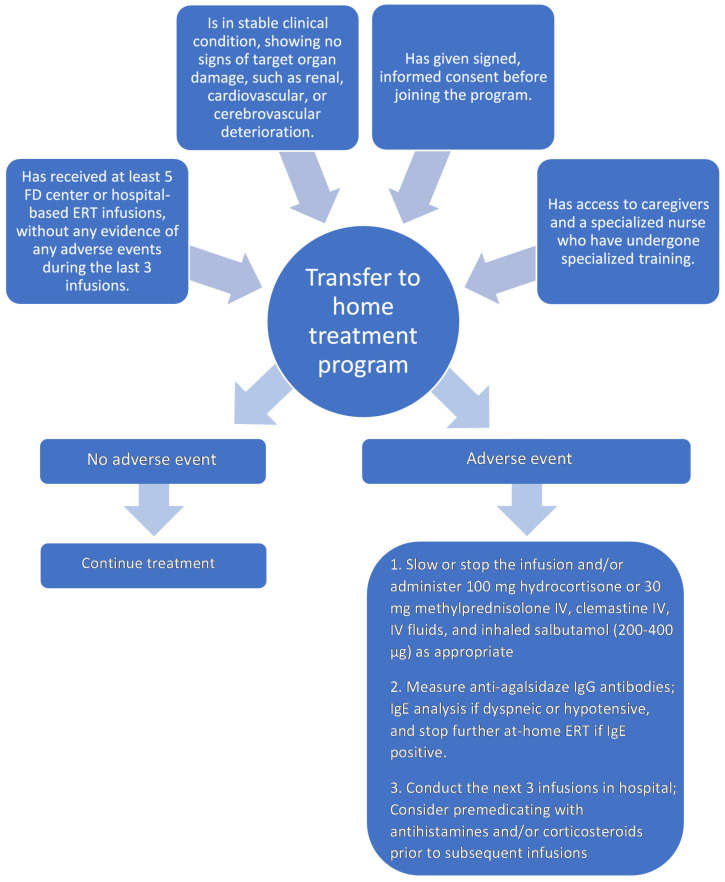
Patient selection process for initiation and continuation of at-home ERT.

**Table 1 ijerph-18-08242-t001:** Availability of home-based ERT programs for Fabry disease in various countries.

Country	Home Infusion Available (Yes/No)
Austria [15]	Y
Argentina [17]	Y
Australia [19]	Y
Canada [20]	Y
Czech Republic [21]	Y
France [20]	Y
Germany [15]	Y
Israel [22]	Y
Italy [7]	Y (11 out of 20 regions)
Netherlands [13]	Y
Norway [14]	Y
Poland	N
Romania [23]	Y
Switzerland [20]	Y
United Kingdom [22]	Y
United States of America [22]	Y

**Table 2 ijerph-18-08242-t002:** Required functions of the mobile application to improve self-monitoring in patients with Fabry disease.

Key Feature of Mobile App	Detailed Features
Secure link to clinic and medical staff	Communication links to selected doctors and nurses
	Direct access to the patient’s clinical data
	Search tool to find the nearest clinic or pharmacy
	Link to pharmacy/drug delivery service for drug orders
Treatment log and patient health records	Personal alerts (e.g., infusion reminders)
	Input drug details (batch number, expiry date), dose, vial serial number, time of infusion
	Input vital signs pre- and post-infusion
	Input pain score, patient-reported outcome survey
	Input supportive treatment used
	Upload clinical photographs (e.g., swollen legs, skin lesions)
	Export data and generate reports
Educational resources	Links to trusted, patient-friendly websites
	Dietary advice and instructions
	Webinars and interactive patient forums

**Table 3 ijerph-18-08242-t003:** Considerations for successful home-based enzyme replacement therapy program implementation.

Benefits
Optimized patient safety during the COVID-19 pandemic by avoiding in-person hospital and clinic visits and transportation to medical centers
Increased patient satisfaction and reduced costs [45]
Good treatment compliance [9]
Increased patient comfort due to regular home visits by experienced medical staff
Convenience of scheduling at-home treatments according to the patients’ daily routine, work, educational commitments, and holidays [17]
**Challenges**
Efficient management and monitoring of infusion-associated events [9]
Potential problems with establishing and maintaining venous access [9]
Requires additional staff willing and able to perform home visits and manage home intravenous ERT infusions of the ERT
Additional costs of the extra staff and their training and supervision
Need for creation of a novel monitoring/alerting system with a medical staff member able to accept patient calls and respond to alerts
Lack of interaction and sharing experiences with other patients [17]

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
