# Peer review of "Considerations for Home-Based Treatment of Fabry Disease in Poland during the COVID-19 Pandemic and Beyond"

_ijerph, 2021, doi:10.3390/ijerph18168242_

Round 1

Reviewer 1 Report

1) The paragraph (r. 67-71) is incomprehensible. It is advisable to rewrite.

2) Table 1. Availability of home-based ERT prog. in various countries - the table is weak. Could you describe yoursource of information in every country you mentioned? And - would it be possible to add the information - % of ERT home - based and/ or number of treated pts in the country vs home based treatment? 

Reviewer 2 Report

The authors give a good review of the status of home treatment for Fabry disease in different countries. It is well structured and reflects the advantages and possible disadvantages that may arise. It demonstrates with many references that in general this type of strategy benefits the patient, improves the quality of life without losing contact with the medical team using different digital solutions and is an important procedure in risk prevention in this pandemic period.

A reference to the coordination phase with the hospital pharmacy team is missing. The home administration procedure requires preparation conditions and a traceability monitoring protocol to ensure that the drug to be infused has followed a controlled process of preparation, temperature and time conditions, which requires an adequate protocol, since patients are generally very far from the central Pharmacy and logistical problems can be solved.
